# The Relevance of Cavity Creation for Several Phenomena Occurring in Water

Giuseppe Graziano

Dipartimento di Scienze e Tecnologie, Università degli Studi del Sannio, Via Francesco de Sanctis, Snc, 82100 Benevento, Italy; graziano@unisannio.it; Tel.: +39-0824-305133

**Abstract:** The solvent-excluded volume effect is an under-appreciated general phenomenon occurring in liquids and playing a fundamental role in many cases. It is quantified and characterized by means of the theoretical concept of cavity creation and its Gibbs free energy cost. The magnitude of the reversible work of cavity creation proves to be particularly large in water, and this fact plays a key role for, among other things, the poor solubility of nonpolar species, the formation of host–guest complexes, and the folding of globular proteins. An analysis of some examples is provided in the present review.

**Keywords:** cavity creation; solvent-excluded volume effect; hydration of noble gases; host–guest complexes; folding of globular proteins

## 1. Introduction

The starting point of any theory able to describe processes occurring in liquids (i.e., pure liquids or solutions) is the recognition that a suitable void space, a cavity, has to be created to allow solute insertion [1–21]. This is the simple consequence of the fact that liquids are a condensed state of matter and each molecule possess its own body. Cavity creation leads to a decrease in the number of configurations accessible to liquid molecules and thus leads to a solvent-excluded volume effect [22–25]. These words are right, but the matter has to be spelled out in more detail to reach a correct understanding (note that cavity creation is a theoretical concept and cannot be studied by performing experiments). Keeping a constant temperature and pressure, the creation of a cavity leads to an increase in liquid volume by the partial molar volume of the cavity itself. This volume increase does not cancel the solvent-excluded volume effect. If the cavity is to exist, the center of liquid molecules (assumed to be spherical) cannot go beyond the solvent-accessible surface area, SASA, and WASA in water [26], of the cavity itself. This means that the shell between the cavity van der Waals surface and the SASA is excluded to liquid molecules, causing a decrease in accessible configurations for basic geometric reasons. The latter constraint does affect the translational motion of all the liquid molecules, not solely the ones in the first solvation shell of the cavity (i.e., of the solute molecule to be hosted).

The solvent-excluded volume effect can be measured by calculating the reversible work of cavity creation, $\Delta G_C$, by means of analytical theories or computer simulations. Classic scaled particle theory, SPT [23–25,27–29], is a simple, geometry-based statistical mechanical model providing analytical formulas to calculate $\Delta G_C$ for cavities of simple shape (i.e., a sphere, a prolate spherocylinder, and others) in liquids made up of hard particles. Its use in the case of water may appear strange, but it works well because the real liquid density is used as input in classic SPT calculations (i.e., density provides indirect information on the strength of the intermolecular attractions existing among liquid molecules; in addition, on the H-bonds between water molecules [30,31]). The classic SPT

formulas to create a spherical cavity in a liquid (neglecting the pressure–volume term for its smallness at P = 1 atm) are:

$$\Delta G_C = RT \cdot \{-\ln(1 - \xi) + [3\xi/(1 - \xi)] \cdot x + [3\xi/(1 - \xi)] \cdot x^2 + [9\xi^2/2(1 - \xi)^2] \cdot x^2\} \quad (1)$$

$$\Delta H_C = [RT^2 \cdot \xi \cdot \alpha_P/(1 - \xi)^3] \cdot [(1 - \xi)^2 + 3(1 - \xi) \cdot x + 3(1 + 2\xi) \cdot x^2] \quad (2)$$

where R is the gas constant, $\alpha_P$ is the isobaric thermal expansion coefficient of the liquid, $\xi$ is the volume packing density of the liquid, which is defined as the ratio of the physical volume of a mole of liquid molecules over the liquid molar volume, $v_1$ (i.e., $\xi = \pi \cdot \sigma_1^3 \cdot N_{Av}/6 \cdot v_1$); $x = \sigma_C/\sigma_1$, and $\sigma_1$ is the hard sphere diameter of liquid molecules; $\sigma_C$ is the cavity diameter, defined as the diameter of the spherical region from which any part of liquid molecules is excluded. The $\Delta G_C$(SPT) magnitude depends upon the volume packing density of the liquid, $\xi$, that is, the fraction of the total liquid volume really occupied by liquid molecules, and the effective hard sphere diameter, $\sigma_1$, of liquid molecules [25,31]. On increasing $\xi$, the void volume decreases and $\Delta G_C$ increases; on decreasing $\sigma_1$, the void volume is partitioned into smaller pieces and $\Delta G_C$ increases (i.e., a significant fraction of the liquid volume is void, but most of these voids are too small to host an atom or a molecule). This implies that the effective diameter of liquid molecules is a fundamental length-scale for the liquid itself. The validity of these arguments has been verified and confirmed in several cases over the years [31].

## 2. Solvation of Noble Gases

A further test is provided here, by analyzing the solvation (i.e., the transfer from a fixed position in the gas phase to a fixed position in the liquid phase) of noble gases in water, carbon tetrachloride, $CCl_4$, and benzene, $C_6H_6$. Experimental thermodynamic data at 25 °C and 1 atm [32–34], reported in Table 1, emphasize that: (1) noble gases are poorly soluble in water, with them being characterized by large positive $\Delta G^.$ values, caused by large negative entropy changes; (2) Ar is characterized by positive $\Delta G^.$ values also in $CCl_4$ and $C_6H_6$, while Xe is characterized by a negative $\Delta G^.$ value in benzene. The $\Delta G_C$(SPT) values calculated for noble gases in the three liquids, whose molecules are assumed to be spherical, at 25 °C and 1 atm, are listed in the eighth column of Table 1. They prove to be largely positive in all cases. Actually, they are significantly larger in water with respect to the other two liquids (see the trends reported in Figure 1).

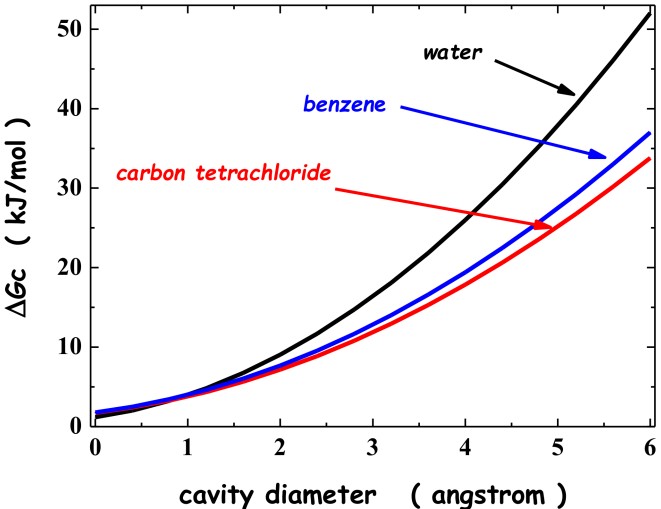

**Figure 1.** Trend of $\Delta G_C$ versus the cavity diameter for water, $CCl_4$, and $C_6H_6$ calculated by means of classic SPT, at 25 °C and 1 atm. The data necessary to perform the calculations are reported in the notes of Table 1.

**Table 1.** Experimental thermodynamic data for the solvation [32–34], according to the Ben–Naim standard (i.e., the transfer from a fixed position in the gas phase to a fixed position in the liquid phase), at 25 °C and 1 atm, of noble gases in water (a), CCl$_4$ (b), and C$_6$H$_6$ (c); the values of the hard sphere diameter and the Lennard–Jones energy parameter come from [35,36] with small modifications; the values of $\Delta G_C$ are calculated by means of classic SPT analytical formulas [27,28]; those of $E_a$ are calculated by means of Pierotti's analytical formula [29].

|   |    | $\sigma$ Å | $\varepsilon/k$ K | $\Delta H^{\cdot}$ kJ mol$^{-1}$ | $\Delta S^{\cdot}$ J K$^{-1}$mol$^{-1}$ | $\Delta G^{\cdot}$ kJ mol$^{-1}$ | $\Delta G_C$ kJ mol$^{-1}$ | $E_a$ kJ mol$^{-1}$ | $\Delta G_C + E_a$ kJ mol$^{-1}$ |
|---|----|-----|-----|------|------|------|------|------|------|
| a | He | 2.6 | 6   | 1.8   | −32.5 | 11.5 | 13.2 | −1.6  | 11.6 |
|   | Ne | 2.8 | 28  | −1.3  | −41.9 | 11.2 | 14.7 | −3.9  | 10.8 |
|   | Ar | 3.4 | 125 | −9.6  | −60.4 | 8.4  | 20.0 | −11.3 | 8.7  |
|   | Kr | 3.7 | 175 | −13.0 | −66.7 | 6.9  | 22.9 | −15.4 | 7.5  |
|   | Xe | 4.0 | 230 | −16.8 | −74.8 | 5.5  | 26.0 | −20.2 | 5.8  |
| b | Ar | 3.4 | 110 | 2.1   | −2.0  | 2.7  | 14.1 | −11.7 | 2.4  |
| c | Ar | 3.4 | 110 | 2.8   | −2.3  | 3.5  | 15.3 | −12.3 | 3.0  |
|   | Kr | 3.7 | 165 | −0.2  | −3.0  | 0.7  | 17.3 | −16.6 | 0.7  |
|   | Xe | 3.4 | 240 | −5.5  | −8.7  | −2.9 | 19.4 | −22.1 | −2.7 |

Additional data used to perform the calculations [37]. Water: $\sigma_1 = 2.8$ Å; $v_1 = 18.07$ cm$^3$·mol$^{-1}$; $\xi = 0.383$; $\alpha_P = 0.257 \cdot 10^{-3}$·K$^{-1}$; $\varepsilon/k = 120$ K. Carbon tetrachloride: $\sigma_1 = 5.37$ Å; $v_1 = 97.09$ cm$^3$·mol$^{-1}$; $\xi = 0.503$; $\alpha_P = 1.226 \cdot 10^{-3}$·K$^{-1}$; $\varepsilon/k = 530$ K. Benzene: $\sigma_1 = 5.26$ Å; $v_1 = 89.41$ cm$^3$·mol$^{-1}$; $\xi = 0.513$; $\alpha_P = 1.22 \cdot 10^{-3}$·K$^{-1}$; $\varepsilon/k = 530$ K.

This holds because, even though the volume packing density of water is the smallest, $\xi = 0.383$ for water versus 0.503 for CCl$_4$, and 0.513 for C$_6$H$_6$, the water molecules are the smallest, $\sigma = 2.80$ Å for water, 5.37 Å for CCl$_4$, and 5.26 Å for C$_6$H$_6$ [35–37]. In this respect, it is important to underscore that the effective hard sphere diameter assigned to water molecules is physically reliable because it corresponds to the location of the first peak in the oxygen–oxygen radial distribution function of water [38], the distance between two H-bonded water molecules. The size effect prevails because it is the molecular cause of the markedly larger number density of water: at 25 °C and 1 atm, $\rho$ (in moles per liter) = 55.3 for water, 10.3 for CCl$_4$, and 11.2 for benzene [37]. The magnitude of the solvent-excluded volume effect associated with cavity creation depends strongly upon the liquid number density: the entropy loss is larger with the greater the number of affected molecules. This is why the size is so important. The reliability of using classic SPT for water has been further confirmed recently by the agreement between the $\Delta G_C$(SPT) values and those obtained by computer simulations in detailed water models [39].

Moreover, a simple formula devised by Pierotti [29] allows for the calculation of the interaction energy $E_a$ between the noble gases and the three liquids. The Pierotti's formula is:

$$E_a = -(64/3) \cdot \xi \cdot \varepsilon_{12} \cdot (\sigma_{12}/\sigma_1)^3 \tag{3}$$

where $\sigma_{12} = (\sigma_1 + \sigma_2)/2$ and $\varepsilon_{12} = (\varepsilon_1 \varepsilon_2)^{1/2}$, and where $\varepsilon_1$ and $\varepsilon_2$ are the Lennard–Jones parameters for the liquid and solute, respectively. The $E_a$ estimates, listed in the ninth column of Table 1, are negative and, when added to the $\Delta G_C$(SPT) values, produce numbers that are close to the experimental $\Delta G^{\cdot}$ ones for all of the three liquids. The success is mainly because the solvent-excluded volume effect associated with solute insertion in a liquid is correctly accounted for by calculating the reversible work of cavity creation.

Estimates of the enthalpy change associated with cavity creation, $\Delta H_C$(SPT), calculated by means of Equation (2) and listed in the third column of Table 2, are positive in all of the three liquids and are close to the values of the $\Delta H^{\cdot} - E_a$ difference, listed in the fourth column of Table 2. This suggests that the structural reorganization of liquid molecules upon noble gas insertion is an endothermic process at 25 °C and 1 atm (i.e., in the case of water, there is no indication of iceberg formation [40–43]). Actually, the $\Delta H_C$(SPT) values of water are significantly smaller than those of the other two liquids [37]; this is a consequence

of the smaller isobaric thermal expansion coefficient αP of water with respect to those of the other two liquids [37] (look at the values reported in the notes of Table 1). The latter quantity, present in the classic SPT formula of $\Delta H_C$ [29], is a measure of the ensemble correlation between fluctuations in volume and fluctuations in enthalpy, and so it can account for the liquid structural reorganization upon cavity creation. The smallness of the $\alpha_P$ of water is due to the strength of water–water H-bonds, in comparison to the weakness of van-der-Waals-type interactions occurring among benzene and carbon tetrachloride molecules [37]. Therefore, the $\Delta H_C$(SPT) values indicate that cavity creation does not cause the breakage of water–water H-bonds [30,31], but a significant breakage of van der Waals interactions occurs in the other two liquids [37].

**Table 2.** Enthalpy and entropy changes associated with cavity creation in water (a), $CCl_4$ (b), and $C_6H_6$ (c), calculated by means of the classic SPT relationships at 25 °C and 1 atm, to be compared with the reorganization enthalpy change and the total solvation entropy change, respectively.

| | | $\Delta H_C$<br>$kJ \cdot mol^{-1}$ | $\Delta H^{\cdot} - E_a$<br>$kJ \cdot mol^{-1}$ | $\Delta S_C$<br>$J \cdot K^{-1} \cdot mol^{-1}$ | $\Delta S^{\cdot}$<br>$J \cdot K^{-1} \cdot mol^{-1}$ |
|---|---|---|---|---|---|
| a | He | 2.1 | 3.4 | −37.2 | −32.5 |
| | Ne | 2.3 | 2.6 | −41.6 | −41.9 |
| | Ar | 3.2 | 2.1 | −56.0 | −60.4 |
| | Kr | 3.7 | 2.4 | −64.1 | −66.7 |
| | Xe | 4.3 | 3.4 | −72.8 | −74.8 |
| b | Ar | 13.4 | 13.8 | −2.3 | −2.0 |
| c | Ar | 14.9 | 15.1 | −1.3 | −2.3 |
| | Kr | 17.1 | 16.4 | −0.7 | −3.0 |
| | Xe | 19.5 | 16.6 | 0.3 | −8.7 |

Estimates of the entropy change associated with cavity creation, $\Delta S_C$(SPT), listed in the fifth column of Table 2, are close to the total solvation entropy changes, listed in the last column of Table 2, in all of the three considered liquids. This finding indicates that the process of cavity creation is the main process responsible of the negative solvation entropy change [31,37]. In water, the $\Delta S_C$(SPT) values are largely negative, increasing in magnitude with the solute diameter [37]. This entropy loss cannot be due to an increase in water structural order [44,45], because it comes from a hard sphere approach. It is due to the decrease in the number of accessible configurations for water molecules because of cavity creation (i.e., the solvent-excluded volume effect). Such a decrease in the number of accessible configurations also occurs in the other two liquids, but it is masked by a largely positive entropy change due to the structural reorganization upon cavity creation [25,37]. The latter structural reorganization, however, has a markedly different magnitude in water and the two organic liquids; it is also characterized by a complete enthalpy-entropy compensation in all liquids [31,46] and does not affect the $\Delta G_C$ magnitude.

## 3. Formation of Host–Guest Complexes

It is interesting that noble gases are able to bind macrocyclic hosts in aqueous solutions. In particular, thanks to specialized NMR experiments, it has been possible to measure the binding constants of noble gases to cucurbit[5]uril, a rigid, synthetic, and water-soluble macrocyclic host [47]. Specifically, at 22 °C, $K_b$(in $M^{-1}$) = 87 for He, 72 for Ne, 360 for Ar, 2390 for Kr and 8700 for Xe. These numbers imply that the binding process is spontaneous, and thus, there is the need to identify the driving force of this host–guest recognition [47–49]. It is important to underscore that the inner part of cucurbit[5]uril proved not to be filled by water molecules, on the basis of both specialized NMR measurements and MD simulations [47] (note that the inner part of cucurbit[5]uril has a volume of 68 $Å^3$ and can host very few water molecules, considering that the van der Waals volume of a water molecule is 11.5 $Å^3$). Researchers calculated with great accuracy, at DFT level, the dispersion energetic attractions of noble gases in bulk water and in the inner part of cucurbit[5]uril. The unexpected result

was that the magnitude of attractive dispersion interactions was larger in bulk water than in the inner part of the rigid macrocyclic host [47]. As a consequence, researchers turned their attention to the reversible work of cavity creation. The transfer of a noble gas atom from water to the inner part of cucurbit[5]uril implies the following steps: the switching-off of the energetic dispersion attractions with water, the closure of the cavity in water, the creation of the cavity in the macrocyclic host interior, and the switching-on of the energetic dispersion attractions with the host.

However, the reversible work to create a cavity in the inner part of cucurbit[5]uril is zero because this region does not contain water molecules (i.e., it is empty); in addition, as a first approximation, the magnitude of the energetic dispersion attractions of a noble gas atom in bulk water and in the host interior can be assumed to be equal. This implies $\Delta G(\text{binding}) \approx -\Delta G_C(H_2O)$, and the driving force is given by the decrease in solvent-excluded volume for cavity closure in water (i.e., leading to a gain in configurational–translational entropy of water molecules).

The experimental $\Delta G(\text{binding})$ values of noble gases to cucurbit[5]uril are reported in Figure 2, together with the values of minus $\Delta G_C(H_2O)$, calculated via classic SPT formulas and listed in Table 1. One could say that the agreement between the two sets of numbers is better than expected, considering their totally different origin. In the original article, the authors calculated $\Delta G_C(H_2O)$ by means of computer simulations, they also considered the contribution of the difference in energetic dispersion attractions between the bulk water and the host interior, and obtained a good agreement with the experimental data [47]. This example demonstrates the pivotal role played by the reversible work of cavity creation in driving host–guest recognition phenomena in water [50–52].

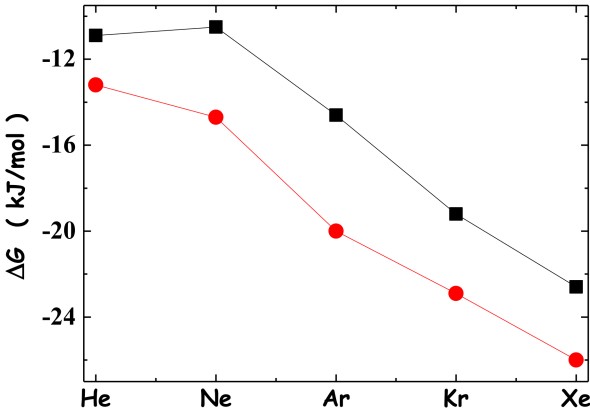

**Figure 2.** Experimental $\Delta G(\text{binding})$ values of noble gases to cucurbit[5]uril, measured via NMR at 22 °C (black filled squares) [47], contrasted with minus the $\Delta G_C$ values for noble gases in water, calculated via classic SPT analytical formulas, and listed in column eight of Table 1 (red filled circles).

## 4. Conformational Stability of Globular Proteins

The geometric explanation for the solvent-excluded volume effect implies that the $\Delta G_C$ magnitude has to increase if the cavity shape is changed, by keeping its van der Waals volume fixed, $V_{vdW}$, and increasing its WASA. This is a fundamental point. Passing from a spherical cavity to several prolate spherocylinders with the same $V_{vdW}$ of the sphere, it is possible to test the rightness of the geometric arguments. Classic SPT analytical formulas allow for the calculation of $\Delta G_C$ for both spherical and prolate spherocylindrical cavities. Therefore, the test can readily be completed and the results have confirmed that $\Delta G_C$ increases with cavity WASA, even though $V_{vdW}$ is fixed [24,53]. This holds true also with $\Delta G_C$ calculated by means of computer simulations [54,55]. The results of classic SPT calculations in water, at 25 °C and 1 atm, for two sets of cavities, the first starting with a sphere of 6 Å radius and the second starting with a sphere of 9 Å radius, are listed in Table 3. It is evident that on lengthening the prolate spherocylinder, WASA increases and $\Delta G_C$ also increases. The plot of $\Delta G_C$ versus WASA, constructed with the numbers of Table 3, is

shown in Figure 3. The $\Delta G_C$ values scale linearly with cavity WASA, but the line slope is not unique; the slope magnitude depends upon the $V_{vdW}$ of the cavity. In fact, the largest spherocylinder of the first set has a WASA larger than that of the smallest spherocylinder of the second set, but the order is reversed in the case of $\Delta G_C$ values (see Table 3 and Figure 3). This means also that the cavity $V_{vdW}$ plays a role [53,56–58].

**Table 3.** $\Delta G_C$ estimates associated with the creation of prolate spherocylindrical cavities, at 25 °C and 1 atm, in a hard sphere fluid, with the experimental density of water and particle diameter $\sigma$ = 2.8 Å. By keeping the cavity $V_{vdW}$ fixed at the volume of 6 Å and 9 Å radius spheres, respectively (i.e., 904.8 Å$^3$ and 3053.6 Å$^3$, respectively), the $\Delta G_C$ numbers have been calculated on varying the cylindrical length by means of the classic SPT analytic formulas. The first row of the A and B sections contains the numbers for the two spherical cavities.

| | $a$ Å | $l$ Å | WASA$_C$ Å$^2$ | $\Delta G_C$ kJ mol$^{-1}$ |
|---|---|---|---|---|
| A | 6.0 | - - | 688.1 | 184.7 |
| | 5.0 | 4.85 | 709.7 | 190.5 |
| | 4.0 | 12.67 | 796.3 | 212.8 |
| | 3.0 | 28.00 | 1017.4 | 266.1 |
| | 2.8 | 33.00 | 1092.5 | 283.4 |
| | 2.5 | 42.75 | 1238.7 | 316.1 |
| | 2.3 | 51.37 | 1366.3 | 343.9 |
| | 2.0 | 69.31 | 1625.9 | 398.3 |
| B | 9.0 | - - | 1359.2 | 399.3 |
| | 7.0 | 10.50 | 1440.9 | 422.9 |
| | 6.0 | 19.00 | 1571.5 | 459.6 |
| | 5.0 | 32.21 | 1810.0 | 524.2 |
| | 4.0 | 55.41 | 2246.5 | 636.9 |
| | 3.5 | 74.69 | 2601.2 | 724.5 |
| | 3.0 | 104.01 | 3118.7 | 847.3 |
| | 2.5 | 152.15 | 3919.5 | 1028.3 |

The geometric formulas for a prolate spherocylinder of radius $a$ and cylindrical length $l$ are: $V_{vdW} = (4/3)\pi \cdot a^3 + \pi \cdot l \cdot a^2$ and WASA $= 4\pi(a + r_w)^2 + 2\pi \cdot l \cdot (a + r_w)$, where $r_w$ is the radius of water molecules, fixed at 1.4 Å; by setting $l = 0$, such formulas become right for a sphere of radius $a$.

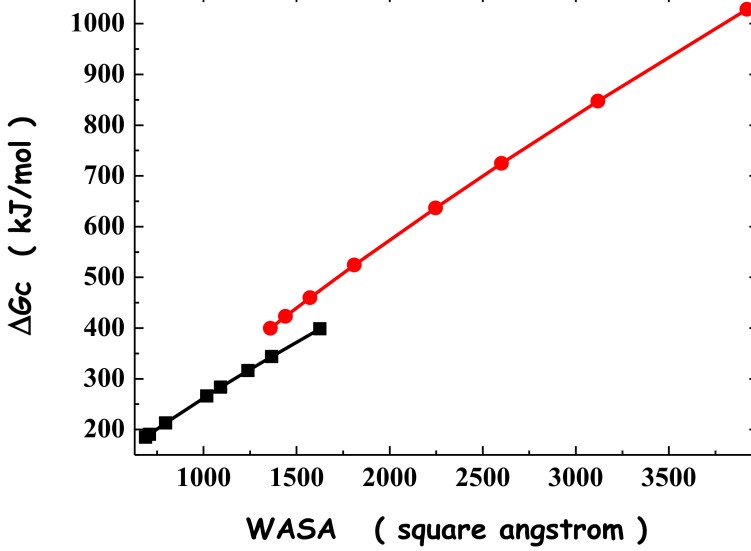

**Figure 3.** Plot of $\Delta G_C$ versus WASA for the two sets of cavities listed in Table 3 (in each set, all of the cavities have the same van der Waals volume). The two lines simply connect the points; they are not the result of a linear regression.

Anyway, the trend of $\Delta G_C$ versus cavity WASA is important to shed light on the driving force of protein folding and on the main factor responsible for the conformational stability of globular proteins. Experimental measurements have proved that the difference in molecular volume between the folded state and the unfolded state ensemble is negligibly small [59,60]. Thus, the folding process can be viewed as a collapse from a set of elongated conformations toward a compact, almost spherical one, keeping the volume occupied by the polypeptide chain fixed [24,25]. Such a collapse is characterized by a large WASA decrease; that means a large $\Delta G_C$ decrease, which corresponds to a significant gain in the configurational–translational entropy of water molecules. The numbers listed in the last column of Table 3 indicate that a large negative Gibbs free energy change is associated, at 25 °C and 1 atm, with the collapse from the longest spherocylinder to the sphere. Polypeptide chains are flexible and can populate different conformations, producing markedly different solvent-excluded volume effects. Water molecules push these chains to assume compact conformations in order to gain configuration–translational entropy. This is the geometric-molecular basis of what is called the hydrophobic effect, considered to be the main determinant of the conformational stability of globular proteins [24,25].

## 5. Conclusions

In the present article, I have tried to show that the solvent-excluded volume effect associated with cavity creation in all liquids (that are a condensed state of the matter) allows one to devise a common and general theoretical approach to rationalize several disparate phenomena occurring in liquids. In particular, the $\Delta G_C$(SPT) values are able to rationalize the low solubility of noble gases in water and its entropic origin, the driving force of the recognition between noble gases and cucurbit[5]uril in water, and, last but not least, a reliable driving force for protein folding and stability.

**Funding:** This research was funded by Università degli Studi del Sannio, FRA 2022.

**Conflicts of Interest:** The author declares no conflict of interest.

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
