# Peer review of "The Relevance of Cavity Creation for Several Phenomena Occurring in Water"

_liquids, doi:10.3390/liquids3010006_

Round 1
Reviewer 1 Report
This is a solid theoretical study in line with previous works from the author. Based on scaled particle theory, the author argues that the excluded volume effect (cavity formation) is a leading driving force in folding and binding reactions. I recommend the manuscript for publication after addressing the following minor issues.
(i) On line 80, the author refers to "Pierottti's analytical formula" to calculate interaction energy, Ea, without showing it. The formula should be given in the revised manuscript.
(ii) It would be appropriate, if the author explains the molecular origins of the enthalpy of cavity formation, DHc, in Table 2. Cavity formation is generally assumed to be a poorly entropic process. Where does enthalpy come from?
(iii) On lines 130-132, the author states that, in non-aqueos solvents, "a decrease in the number of accessible configurations... is masked by by the large positive change due to the structural reorganization upon cavity creation". The authors should explain the origin of the structural reorganization, the positive change in entropy it causes, and why a similiar "masking" does not occur in water.
(iv) Lines 190-194. The sentence "The DGc values scale linearly..." is confusing and should be rephrased to make it easier to follow.
Author Response
I thank you the Reviewer for the positive evaluation of the manuscript and for the useful suggestions.
I have inserted the Pierotti's formula to calculate the solute-solvent interaction energy; I have discussed in more detail the structural reorganization of liquid molecules upon cavity creation that is the cause of the total enthalpy change and of part of the entropy change; I have slightly modified the sentence for the linear scaling between the work of cavity creation and cavity SASA.
Reviewer 2 Report
In the manuscript “The Relevance of Cavity Creation for Several Phenomena Occurring in Water”, Graziano presents an interesting contribution to the investigation of molecular aspects of solvation, with implications for aqueous solubility, host-guest complexes and folding of globular proteins. His main goal is to understand the free energy of cavity creation, its decomposition in entropic and enthalpic terms and the interpretation of these terms. The entropy loss in the cavity creation is explained by the decrease in the number of configurations accessible to liquid molecules due to the solvent-excluded volume effect rather than due to the increase of water structural order. To support this hypothesis (already discussed in the literature) the author uses classic scaled particle theory to calculate the free energy of cavity creation in some systems, aqueous and non-aqueous. The author found a very good agreement between the experimental free energy of solvation and the calculated free energy of cavity creation summed with the interaction energy. Besides, the enthalpic and entropic contributions were successfully explained, for both aqueous and non-aqueous systems. Moreover, the free energy of cavity creation was found to be dependent both on the shape of cavity and on the van der Waals volume of the cavity. The findings were discussed to explain the cavity closure in water as the driving force for the formation of host-guest complexes and also the decrease of the water-accessible surface area as the driving force of the folding of globular proteins.
Despite the results being interesting and relevant and considering the relevance of the interpretation, I find the author could improve the manuscript showing the equations used to calculate the quantities (cavitation free energy, cavitation enthalpy and entropy, interaction energy) and describing the methodological details of these calculations in the main text or in a supplementary material.
Author Response
I thank the Reviewer for the positive comments on the manuscript and the suggestions to improve it.
Following the indications, I have added to the main text three Equations corresponding to the classic SPT relationships for the work of cavity creation and the associated enthalpy change and the Pierotti's formula to calculate the solute-solvent interaction energy. I have also added the relevant explanations.